# Cytokine Regulation and Oxidative Stress in Helicobacter Pylori-Associated Gastric Adenocarcinoma at Different Stages: Insights from a Cross-Sectional Study

**DOI:** 10.3390/ijms26157609

**Published:** 2025-08-06

**Authors:** Olga Smirnova, Aleksander Sinyakov, Eduard Kasparov

**Affiliations:** Federal Research Center “Krasnoyarsk Science Center”, Siberian Branch of the Russian Academy of Sciences “Scientific Research Institute of Medical Problems of the North”, St. Partizan Zheleznyaka, 3 “G”, 660022 Krasnoyarsk, Russia; sinyakov.alekzandr@mail.ru (A.S.); impn@impn.ru (E.K.)

**Keywords:** gastric adenocarcinoma, *Helicobacter pylori*, cytokines, oxidative stress

## Abstract

Gastric adenocarcinoma is a malignant tumor that develops from the glandular cells of the inner wall of the stomach. The prevalence of this type of disease varies from 90 to 95% of all types of gastric cancer. The aim of our study was to investigate the differences in the content of cytokines and oxidative stress markers in patients with gastric adenocarcinoma associated with *H. pylori* infection depending on the stage. The study included 281 patients with gastric cancer. At stage I of the disease—75 people, stage II—70 people, stage III—69 people, and stage IV of the disease—67 people. The levels of TNF-α, IL-2, IL-8, IFNγ, TNF-β, IL-17A, IL-6, IL-10, and IL-4 in the blood serum of patients and healthy individuals were determined by enzyme immunoassay and plasma oxidative stress scores (MDA, SOD, CAT, GST, GPO, CP). The present study revealed that *H. pylori*-infected gastric adenocarcinoma at different stages is associated with different plasma levels of cytokines, lipid peroxidation products, and antioxidant defense factors. Further studies are needed to evaluate the effectiveness of therapeutic strategies combining cytokine regulation and oxidative stress to improve clinical outcomes in gastric cancer.

## 1. Introduction

Gastric adenocarcinoma is considered one of the most common malignant neoplasms with a high mortality rate [1,2]. One of the reasons for the high mortality in this type of disease is the aggressive course of the pathological process with the development of distant metastases. Metastatic lesions in gastric cancer can develop through lymphogenous, hematogenous, and implantation routes. In this case, metastatic lesions of the abdominal cavity, including the peritoneum and greater omentum, occur in more than 55–60% of patients [3,4].

The state of the immune system plays an important role in tumor growth and dissemination. According to modern concepts, the interaction of a malignant tumor and a tumor-bearing organism has a number of successive phases: elimination of tumor cells under the influence of innate and adaptive immunity factors, equilibrium with the formation of clones of dominant tumor cells, and the evasion of tumor cells from the action of immune system cells, accompanied by the formation of a growth-stimulating and immunosuppressive microenvironment [5,6,7]. An integral part of the latter are cytokines and growth factors.

Cytokines are molecules produced by activated immunocompetent and some other cells, in particular tumor and endothelial cells. Many of them can act as factors enhancing tumor progression, activating angiogenesis (IL-1ß, TNF-α), tumor cell migration (IL-1ß, IL-6, IL-8), epithelial cell metaplasia (IL-6), and some stimulators and products of T regs (IL-10) [8,9]. At the same time, they are also capable of exhibiting anti-oncogenic effects: for example IL-1ß, IL-6, IL-8—by involving various links of the immune system in the antitumor response; TNF-α—being an apoptosis inducer; and IL-10—due to antagonism in relation to pro-inflammatory interleukins and increasing the production of toxic molecules in the tumor (for example, nitric oxide NO), which prevent the growth of the neoplasm [10].

With the development of malignant diseases, changes occur in the systems “lipid peroxidation (LPO)—antioxidant protection (AP)”. These changes become pathogenetic and contribute to tumor progression and the development of metastases. Oxidative degradation of lipids that occurs during membrane lipid peroxidation occurs under the influence of free radicals, and the consequences of LPO are balanced by the antioxidant protection system. However, when the balance of AP is disturbed, oxidative stress occurs, triggering pathological modification of the body’s cells and their structures. The system of tissue and cell protection from toxic oxygen metabolites and LPO products can be conditionally divided into physiological (mechanisms that regulate the delivery and supply of oxygen to cells) and biochemical (the AP of the body itself, which includes a wide class of chemical compounds that reduce the activity of radical oxidative processes). The physiological component of the AP system of the body ensures a balance between the intensity of oxygen transport to cells and metabolic processes for its beneficial and safe utilization. Despite a sufficient number of studies of LPO-AP, the role of these systems in tumor development has not been fully determined [11].

Currently, combined and complex treatment methods are proposed for the treatment of gastric cancer [12]. However, to improve both the treatment and early diagnosis of gastric adenocarcinoma, a comprehensive understanding of the pathogenetic processes of the biological behavior of the tumor is necessary [13,14]. This approach requires fundamental research, which determines the relevance of our study. The aim of our study was to examine the differences in the content of cytokines and oxidative stress markers in patients with gastric adenocarcinoma associated with *H. pylori* infection depending on the stage.

## 2. Results

### 2.1. Baseline Patient Characteristics

The mean age of the 281 patients and comparison group was 54.3 years (±9.4). The gastric cancer group included one hundred and sixty-nine patients, divided by the stage of disease. The control and study groups were similar in baseline demographic data (Table 1).

### 2.2. Study of the Cytokine Link in Patients with Gastric Cancer at Different Stages of the Disease

When studying the cytokine link in patients with gastric cancer at different stages of the disease, we found an increase in TNF-α in patients at stages I, II, and III of the disease compared to the control group and the group of patients at stage IV of the disease (p_1–2_ < 0.001, p_1–3_ < 0.001, p_1–4_ < 0.001) (Table 2). When studying stage IV gastric cancer, a 10-fold decrease in TNF-α was found compared to the control group (p_1–5_ < 0.001; p_2–5_ < 0.001; p_3–5_ < 0.001; p_4–5_ < 0.001).

When studying IL-2, an increase in this cytokine was found in all study groups compared to the control (p_1–2_ < 0.001, p_1–3_ < 0.001, p_1–4_ < 0.001, p_1–5_ < 0.001).

In patients with gastric cancer of stages I, II, and III of the disease, an increase in IL-8 was observed compared to the control group and the group of patients at IV (p_1–2_ < 0.001, p_1–3_ < 0.001, p_1–4_ < 0.001). At the same time, this cytokine decreased 5 times in patients with gastric cancer of stage IV compared to the control group (p_1–5_ = 0.01, p_2–5_ < 0.001, p_3–5_ < 0.001, p_4–5_ < 0.001). Our study revealed a decrease in IFNγ in patients with gastric cancer at stages I, II, and III of the disease compared with the control group and the group of patients at stage IV (p_1–2_ < 0.001, p_1–3_ < 0.001, p_1–4_ < 0.001, p_2–4_ = 0.02).

In patients with stage IV disease, there was an increase in IFNγ compared to the control group (p_1–5_ < 0.001, p_2–5_ < 0.001, p_3–5_ < 0.001, p_4–5_ < 0.001).

In the study of TNF-β, an increase in this indicator was found in patients with gastric cancer at stages I, II, and III compared to the control group and the group of patients at stage IV of the disease (p_1–2_ < 0.001, p_1–3_ < 0.001, p_1–4_ < 0.001). In the study of stage IV gastric cancer, a 10-fold decrease in TNF-β was found compared to the control group (p_1–5_ < 0.001, p_2–5_ < 0.001, p_3–5_ < 0.001, p_4–5_ < 0.001).

In all study groups, there was an increase in IL-17A and multiple increases in IL-6 and IL-10 in all study groups compared to the control (IL-6: p_1–2_ < 0.001, p_1–3_ < 0.001, p_1–4_ < 0.001, p_1–5_ < 0.001; IL-10: p_1–2_ < 0.001, p_1–3_ < 0.001, p_1–4_ < 0.001, p_1–5_ < 0.001).

When studying IL-4, an increase in this indicator was found in patients with gastric cancer at stages I, II, and III of the disease compared to the control group and the group of patients at stage IV of the disease (p_1–2_ < 0.001, p_1–3_ < 0.001, p_1–4_ < 0.001). When studying this cytokine in patients with stage IV gastric cancer, a decrease in IL-4 was found compared to the control group (p_1–5_ = 0.03, p_2–5_ < 0.001, p_3–5_ < 0.001, p_4–5_ < 0.001).

In patients with gastric cancer at stages I, II, and III of the disease, there is an increase in such proinflammatory cytokines as TNF-α, IL-2, IL-8, TNF-β, IL-17A, and IL-6, as well as a decrease in IFNγ. When studying anti-inflammatory cytokines, an increase in IL-10 and IL-4 was found. Thus, in patients at stages I, II, and III of the disease, an immune response occurs, predominantly by the Th2/Th17 mechanism.

In patients with gastric cancer at stage IV of the disease, multidirectional changes occur when studying proinflammatory cytokines: a decrease in TNF-α, IL-8, TNF-β, as well as an increase in IL-2, IFNγ, IL-17A, and IL-6. The pronounced cytokine imbalance is probably due to a change in the functions of immune system cells from antitumor to protumor.

### 2.3. Indicators of LPO–AP in Gastric Cancer Depending on the Stage of the Disease

At the next stage, we studied the parameters of the LPO-AOD system in the plasma of patients with gastric cancer at different stages of the disease. When studying the lipid peroxidation system, the concentrations of secondary lipid peroxidation products (malonic dialdehyde) formed at the stages of the free radical chain reaction were investigated. The activity of antioxidant protection was judged by the content of its main components (superoxide dismutase, catalase, glutathione-S-transferase, glutathione peroxidase, ceruloplasmin) (Table 3).

The study found an increase in the median MDA in patients at all stages of the disease relative to the control group (MDA: p_1–2_ < 0.001, p_1–3_ < 0.001, p_1–4_ < 0.001, p_1–5_ < 0.001; p_2–5_ = 0.03) (Table 3). Next, the state of the AOP system in the patient groups was assessed. It was revealed that the median values of glutathione-S-transferase, glutathione peroxidase, and ceruloplasmin in plasma increased in all groups of patients compared to the control group (GST: p_1–2_ < 0.001; p_1–3_ < 0.001; p_1–4_ < 0.001; p_1–5_ < 0.001, GPO: p_1–2_ < 0.001; p_1–3_ < 0.001; p_1–4_ < 0.001; p_2–4_ = 0.03; p_3–4_ = 0.03; p_1–5_ < 0.001; p_2–5_ = 0.03; p_3–5_ = 0.03, CP: p_1–2_ < 0.001; p_1–3_ < 0.001; p_1–4_ < 0.001, p_1–5_ < 0.001). The study of superoxide dismutase activity revealed an increase in this indicator in the groups of patients at stages I and II compared to the control (SOD: p_1–2_ < 0.001; p_1–3_ < 0.001) and a decrease in the activity of this enzyme in patients with gastric cancer at stages III and IV compared to the control and with the groups of patients at stages I and II (p_1–4_ = 0.03; p_1–5_ < 0.001). The study of CAT revealed a decrease in the activity of this enzyme in all studied groups relative to the control (p_1–2_ = 0.04, p_1–3_ = 0.04, p_1–4_ = 0.04, p_1–5_ = 0.003).

The oxidative stress coefficient indicates the equilibrium between lipid peroxidation processes and the antioxidant system, typically aiming for 1. In patients with gastric cancer, the oxidative stress coefficient was measured, revealing that at stage I it was 16.6, at stage II it rose to 17, at stage III it further increased to 34.1, and at stage IV it reached 56.5.

In patients with gastric cancer at stages I and II of the disease, there is an increase in the concentration of the end products of lipid peroxidation—MDA, as well as an increase in the activity of such enzymes of antioxidant protection as SOD, GST, GPO, and CP, and a decrease in the activity of CAT.

In patients with gastric cancer at stages III and IV of the disease, there is an increase in the concentration of the end products of lipid peroxidation—MDA, as well as an increase in the activity of such enzymes of antioxidant protection as GST, GPO, and CP, and a decrease in the activity of the enzymes SOD and CAT.

### 2.4. Regression Analysis of the Studied Cytokines and LPO-AOP Indices in Gastric Cancer

In a simple linear regression analysis study in patients with gastric cancer at stages I, II, III, and IV, the following results were obtained. A simple linear regression analysis performed among all participants showed that MDA was significantly correlated with IFNγ (Rs = 0.71; *p* < 0.001), IL-4 (Rs = 0.587; *p* < 0.01), and SOD (Rs = 0.66; *p* < 0.02). In addition, SOD was significantly correlated with IFNγ (Rs = 0.709; *p* < 0.01) and IL-4 (Rs = 0.73; *p* < 0.001).

To identify independent predictors of MDA in the general population, multiple linear regression analyses were performed including variables linearly associated with the dependent variable, such as IFNγ, IL-4, and SOD, depending on the disease stage. IFN (SE: 0.012; standardized β coefficient: 0.589; *p* < 0.01) and SOD (SE: 0.642; standardized β coefficient: 0.296; *p* = 0.01) were the only independent predictors of MDA (R^2^ = 82%).

## 3. Discussion

The lipid peroxidation–antioxidant defense system is more sensitive and responds more quickly to the growth of gastric adenocarcinoma tumors infected with *H. pylori*.

An increase in the content of ceruloplasmin, a protein with ferroxidase activity, occurs at all stages of the disease, indicating the involvement of iron metabolism in the pathogenesis of gastric cancer infected with *H. pylori*. Ceruloplasmin, a copper-containing glycoprotein that exhibits ferroxidase and superoxide-scavenging activities, is one of the most important antioxidant proteins in blood plasma [15]. This protein inhibits superoxide and ferritin-dependent lipid peroxidation in lipoprotein particles of blood plasma.

The decrease in catalase activity at all stages of gastric cancer infected with *H. pylori* is probably due to substrate inhibition by excess hydrogen peroxide, which, on the other hand, causes an increase in the activity of the enzyme glutathione peroxidase for its neutralization [16]. Glutathione peroxidase is the most important enzyme for neutralizing the effects of lipid peroxidation. This group of enzymes actively participates in the processes of protecting cells and tissues of the body from the damaging toxic effects of metabolic products. This is especially true for lipid peroxidation. Despite the numerous positive processes occurring during LPO, such as apoptosis, regulation of the transmembrane transfer of biologically active substances, enzymes, receptors, and much more, there is also a negative role for these processes: for example, the ability to damage cell membranes, disrupting their structure, and maintaining cell homeostasis. Glutathione peroxidase ensures the conversion of hydrogen peroxide and its radical derivatives into harmless compounds [17,18]. Catalase, a heme-containing enzyme of the oxidoreductase class, whose main function is the ability to catalyze the reaction of decomposition of hydrogen peroxide (H_2_O_2_) into water and molecular oxygen (O_2_), protects cells from destruction by active oxygen species. Catalase does not have an extracellular isoform; it enters the blood plasma along with cellular contents as a result of the breakdown of cells and tissues.

Oxidative stress develops at all (even early) stages of gastric cancer infected with *H. pylori*. Malondialdehyde is a product of lipid peroxidation and serves as a marker of radical oxidation processes triggered in cells by active oxygen species. As a result of further biochemical transformations, it is oxidized to carbon dioxide or interacts with phospholipids, amino acids, and nucleic acids. Currently, malondialdehyde is considered as a marker of oxidative stress [19]. High levels of MDA are detected in many diseases, including gastric cancer. On this basis, MDA can be used as an auxiliary prognostic marker in the examination of patients with gastric adenocarcinoma. Increased MDA levels in plasma indicate excessive production of ROS and do not exclude significant damage to the endothelium of blood vessels in patients with gastric cancer.

As the tumor develops, the immune response changes from Th2/Th17 to Th1 with an imbalance in cytokine levels. IFNγ is considered one of the most important proinflammatory cytokines, produced by activated T lymphocytes. This cytokine is an important activator of macrophages and an inducer of the expression of class II major histocompatibility complex (MHC) molecules. IFNγ is produced primarily by NK and NKT cells as part of the innate immune response, as well as by effector T cells CD4 Th1 and CD8 cytotoxic T lymphocytes after the development of antigen-specific immunity as part of the adaptive immune response. IFNγ production by natural killers is triggered by their interaction with target cells (tumor, virus-infected). The main manifestations of the biological activity of TNF-α are as follows: selective cytotoxicity against some tumor cells, inhibition of the synthesis of the key enzyme of lipogenesis-lipoprotein kinase, participation in the regulation of the immune response and inflammation. This interleukin performs the most important functions during the onset of inflammation. TNF-α activates the endothelium, promotes the adhesion of leukocytes to the endothelium by inducing the expression of adhesion molecules on endothelial cells and the subsequent transendothelial migration of leukocytes to the site of inflammation, and, in addition, activates leukocytes (granulocytes, monocytes, lymphocytes) and induces the production of other proinflammatory cytokines: IL-1, IL-6, IFNp, GM-CSF, which have a synergistic effect with TNF-a [20,21].

Cytokine imbalance in stage IV gastric cancer infected with *H. pylori* is likely due to changes in the functional activity of immune cells. Regression analysis performed among all participants showed that MDA was significantly correlated with IFNγ, IL-4, and SOD. To identify independent predictors of MDA in the general population, multiple linear regression analyses were performed including variables linearly related to the dependent variable, such as IFNγ, IL-4, and SOD, depending on the stage of the disease.

## 4. Materials and Methods

### 4.1. Subjects

This study included 281 patients with stomach cancer. The first group consisted of patients with stomach cancer T1N0M0—stage I—75 people (45 to 59 years old, average age 49 ± 4.7). The second group consisted of patients with stomach cancer T2N0M0—stage II—70 people (45 to 59 years old, average age 53 ± 5.7). The third group consisted of patients with stomach cancer T3N0M0—stage III—69 people (45 to 59 years old, average age 55 ± 3.7). The fourth group consisted of patients with stomach cancer T4N0M0—stage IV—67 people (45 to 59 years old, average age 54 ± 5.8). The clinical assessment of individuals with gastric cancer was performed at the Krasnoyarsk Regional Clinical Oncology Dispensary named after A.I. Kryzhanovsky, in the Oncoabdominal Surgery Department named after N.A. Rykovanov (Figure 1). An oncologist established the diagnosis utilizing clinical, anamnestic, laboratory, and instrumental information. Participants were enrolled in the study, and biological samples were collected upon hospital admission prior to the initiation of treatment. The sample used for the study was venous blood collected from the patient in the morning between 8 and 9 am while fasting from the cubital vein, using Vacutainer tubes that contained a separating gel and a dual coagulation activator (silica), as well as a Vacutainer with sodium heparin solution (5 U/mL). The control group consisted of 100 mostly healthy blood donors with an average age of (48.7 ± 3.9 years), who had no gastrointestinal complaints or history and showed no changes in the gastric mucosa. The research excluded patients with HIV; those experiencing hepatitis, tuberculosis, gastric ulcers; or those with concurrent acute and chronic illnesses in the exacerbation stage. The research did not involve patients who declined to take part in the scientific study.

The study was conducted with the permission of the Ethics Committee of the Research Federal Research Center “Krasnoyarsk Science Center” of the Siberian Branch of the Russian Academy of Sciences (Protocol No. 7 dated 9 September 2024) and with the permission of the Ethics Committee of the Krasnoyarsk Regional Clinical Oncology Dispensary named after A.I. Kryzhanovsky. This study was approved by the local Ethics Committee of the Krasnoyarsk Regional Clinical Oncology Dispensary. In working with the examined patients, the ethical principles presented in the Declaration of Helsinki of the World Medical Association were observed. All participants were informed about the purpose and procedure of the study, and they were asked to give written informed consent for the examination and study.

### 4.2. Endoscopic Examination, Histologic Examination, H. pylori Test, and Gastric Juice Sampling

All participants in the study received three forms of testing for *Helicobacter pylori*: histological analysis, quick urease test, and culture. All biopsies were collected by a skilled oncologist. To prevent contamination, the endoscope underwent washing and disinfection through immersion in a detergent solution. Moreover, serum IgG to *H. pylori* was assessed using an enzyme immunoassay (BIOHIT HealthCare, Helsinki, Finland). Antibody titers of 30 EIU or higher were regarded as positive, whereas titers below 30 EIU were deemed negative for *H. pylori*. All participants in the research demonstrated current infection with *H. pylori*: histological confirmation of *H. pylori* infection via modified Giemsa staining, a positive urease test (CLOtest; Delta West, Bentley, Western Australia), a positive culture test for *H. pylori*, and an ELISA technique assessing IgG levels to *H. pylori* in serum.

### 4.3. Determination of Cytokines by ELISA

The levels of TNF-α, IL-2, IL-8, IFNγ, TNF-β, IL-17A, IL-6, IL-10, IL-4 in the blood serum of patients and healthy individuals were determined by enzyme immunoassay on a Muitiskan FC analyzer (ThermoFisher Scientific Inc., Waltham, MA, USA) using reagent kits manufactured by JSC “Vector-Best” (Novosibirsk, Russia).

The solid-phase immunoassay method used is based on the “sandwich” principle. The analysis is performed in two stages. In the first stage, calibration samples with a known concentration of cytokines are used, and the samples under study are incubated in the wells of a strip plate with immobilized monoclonal antibodies (MAB) to certain interleukins. Then, the plate is washed. In the second stage, the cytokines bound in the wells are treated with a MAB conjugate to cytokines with peroxidase (the MAB conjugate and the MAB immobilized in the wells of the plate are specific to different areas). After washing away the excess conjugate, the resulting immune complexes “immobilized MAB-cytokine -conjugate” are detected by an enzymatic reaction of peroxidase with hydrogen peroxide in the presence of a chromogen (orthophenylenediamine). The intensity of the chromogen color is proportional to the concentration of the cytokine in the analyzed sample. After stopping the peroxidase reaction with a stop reagent, the results are taken into account photometrically. The concentration of cytokines in the samples is determined using a calibration graph.

### 4.4. Lipid Peroxidation and Antioxidants

Scores of oxidative stress in plasma (MDA, SOD, CAT, GST, GPO, CP). The indicators of lipid peroxidation and the antioxidant system were assessed using the blood plasma of the subjects through the spectrophotometric method on the Thermo SCIENTIFIC GENESYS 10vis device (Thermo Fisher Scientific, USA).

### 4.5. Determination of Malondialdehyde Content

MDA is formed in lipids as a result of their peroxidation. When interacting with 2-thiobarbituric acid (TBA), a chromogen is formed with an absorption maximum in the red region of the visible spectrum at a wavelength of 532 nm. The MDA content is calculated taking into account the molar extinction coefficient of the formed chromogen, equal to 1.56 105 M^−1^ cm^−1^, and is expressed in μmol/L:$$C=\frac{D532\times V_{p.c.}\times 1000}{V_{np}\times \varepsilon \times B\times d}$$

### 4.6. Determination of Superoxide Dismutase Activity

The method’s principle involves the suppression of adrenaline autooxidation reactions in an alkaline environment with superoxide dismutase (SOD) present. The intensity of adrenaline autooxidation was assessed by the dynamic rise in absorption at a wavelength of 347 nm, resulting from the buildup of the oxidation product and the creation of adrenochrome, which has an absorption peak at 480 nm, occurring prematurely:$$Unit\;of\;activity\frac{SOD}{gPr}=(\frac{Ex-Eo}{Ex})*\;\frac{100\%*F\times V\times 1000}{50\times v\times d\times C}$$
where

$\frac{Ex-Eo}{Ex}\;\times \;\frac{100\%}{50}$—unit of activity per mL of plasma.

### 4.7. Determination of Catalase Activity

The measurement of catalase activity relies on the creation of a yellow complex formed by hydrogen peroxide and ammonium molybdate, which remains intact throughout the catalase reaction. Catalase activity is determined using the equation:$$A=\frac{\frac{\Delta A}{c}\times V\times f}{t\times v\times d\times K\times 60}$$

### 4.8. Determination of Glutathione-S-Transferase Activity

The activity of glutathione-S-transferase was assessed by the speed of producing glutathione-S-conjugates from GSH and 1-chloro-2,4-dinitrobenzene (CDNB). The rise in conjugate concentration throughout the reaction was measured spectrophotometrically at 340 nm (glutathione-S-CDNB’s peak absorption). The enzyme activity was determined using the millimolar extinction coefficient for GS-CDNB at a wavelength of 340 nm, which is 9.6 mM^−1^ cm^−1^, and reported as micromoles of glutathione-S-conjugates produced per minute per 1 g of protein:$$A=\frac{\Delta E/\min\times V_{p.c.}\times 1000}{\varepsilon \times V_{n.}\times d}$$

### 4.9. Determination of Glutathione Peroxidase Activity

Glutathione peroxidase (GPO) facilitates the interaction between glutathione (GSH) and tert-butyl hydroperoxide (TBH):


*GSH + TBH = (GPO) = GSSO + restoredTBH.*


The enzyme activity was evaluated by measuring the variation in GSH levels in the samples prior to and following incubation with the model substrate during a color reaction involving dithionitro(bis)benzoic acid (DTNBK). The extinction of both experimental and control samples was quantified using a spectrophotometer set to a wavelength of 412 nm, calibrated with purified water.

Activity is calculated using the formula$$A=\frac{\Delta D\times V_{p.c.}\times 1000}{V_{np}\times \varepsilon \times d},$$

### 4.10. Determination of Oxidative Stress Index (OSR)

Technique for personal evaluation of oxidative stress through the computation of the integral coefficient derived from the proportion of pro- and antioxidant factors, where the indicator levels of the patients are being studied; n represents the indicator levels of the comparison group; with OSR > 1, oxidative stress development is noted.

OSR calculation formula:$$OSR=\frac{MDA\;\left(i\right)/MDA\left(n\right)}{\left(\frac{SOD\left(i\right)}{SOD\left(n\right)}\right)\times \left(\frac{CAT\left(i\right)}{CAT\left(n\right)}\right)\times \left(\frac{GST\left(i\right)}{GST\left(n\right)}\right)\times \left(\frac{GPO\left(i\right)}{GPO\left(n\right)}\right)}$$

### 4.11. Statistical Analysis

Statistical data processing was performed using the Statistica 7.0 software package (Statistica. Dell StatSoft, Tulsa, OK, USA). The analysis of the conformity of the distribution type of the feature to the normal distribution law was performed using the Shapiro–Wilk criterion. When describing the sample, medians (Me) and the interquartile range of percentiles (C_25_–C_75_) were calculated. The reliability of differences between the indicators of independent samples was estimated using the Mann–Whitney criterion (*p* < 0.05). Multivariate regression analysis was performed using two predictors. According to Green’s rule [22], a minimum sample size of 66 subjects (N ≥ 50 + 8 × m, where m = number of predictors) is required to detect medium effect sizes when testing the general model. Given our sample size, the analysis meets this recommended criterion.

## 5. Conclusions

In conclusion, the present study revealed that *H. pylori*-infected gastric adenocarcinoma at different stages is associated with different plasma levels of cytokines, lipid peroxidation products, and antioxidant defense factors. Further studies are needed to evaluate the effectiveness of therapeutic strategies combining cytokine regulation and oxidative stress to improve clinical outcomes in gastric cancer.

## Figures and Tables

**Figure 1 ijms-26-07609-f001:**
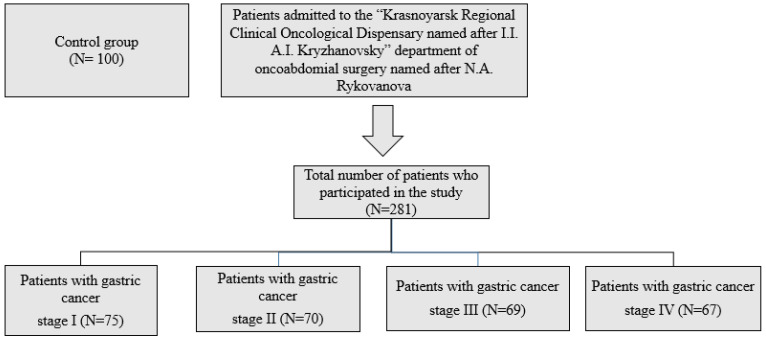
Scheme of inclusion of patients in the study.

**Table 1 ijms-26-07609-t001:** Initial demographic data evaluation of two groups.

Parameter	Control Group N = 100 (1)	Stomach Cancer Stage I N = 75 (2)	Stomach Cancer Stage II N = 70 (3)	Stomach Cancer Stage III N = 69 (4)	Stomach Cancer Stage IV N = 67 (5)	*p*-Value
Gender (n,%) Male Female						0.325
	58	35	26	22	18	
	42	14	19	18	17	
Age, (y)	47.54 ± 12.65	49.3 ± 9.65	53.87 ± 8.34	55.4 ± 6.34	56.1 ± 4.34	0.414
Weight, (kg)	58.4 (±5.3)	56.1 (±3.2)	53.2 (±4.1)	57.4 (±4.3)	54.3 (±3.2)	0.256
Height, (m)	1.75 (±0.05)	1.7 (±0.08)	1.69 (±0.05)	1.72 (±0.09)	1.67 (±0.08)	0.465
BMI, (kg/m^2^)	22.69 ± 3.71	19.52 ± 3.33	17.22 ± 3.89	18.52 ± 4.2	16.32 ± 3.9	0.398
Alcohol drink 1. Never 2. Past 3. Current						0.472
	16 (15.3)	5 (10.9)	7 (13.8)	10 (13.1)	5 (9.2)	
	41 (17.4)	20 (20.7)	23 (11.8)	16 (12.8)	20 (23.1)	
	43 (71.7)	24 (65.5)	15 (82.4)	14 (71.3)	10 (14.3)	
Smoking 1. Never 2. Past 3. Current						0.371
	50 (44.1)	12 (18.3)	11 (33,1)	16 (27.1)	13 (23.2)	
	18 (43.5)	7 (47.1)	11 (37.9)	14 (25.2)	10 (30.8)	
	32 (21.7)	20 (23.5)	8 (20.7)	10 (19.4)	12 (21.7)	

**Table 2 ijms-26-07609-t002:** Quantitative content of cytokines in patients with gastric cancer at different stages of the disease (Me, C_25_–C_75_, p_m-u_).

Indicators	Control Group N = 100 (1)		Stomach Cancer Stage I N = 75 (2)		Stomach Cancer Stage II N = 70 (3)		Stomach Cancer Stage III N = 69 (4)		Stomach Cancer Stage IV N = 67 (5)	
	Me	C_25_–C_75_	Me	C_25_–C_75_	Me	C_25_–C_75_	Me	C_25_–C_75_	Me	C_25_–C_75_
TNF-α (pg/mL)	3.54	0.8–7.87	43.2	40.1–60.9	48.9	37.4–55.7	38.1	32.3–43.4	0.3	0.1–0.4
			p_1–2_ < 0.001		p_1–3_ < 0.001		p_1–4_ < 0.001		p_1–5_ < 0.001; p_2–5_ < 0.001; p_3–5_ < 0.001; p_4–5_ < 0.001	
IL-2 (pg/mL)	1.1	0.5–3.05	5.7	3.6–10.3	4.9	3.8–9.5	5.2	3.0–8.7	8.4	7.3–15.4
			p_1–2_ < 0.001		p_1–3_ < 0.001		p_1–4_ < 0.001		p_1–5_ < 0.001	
IL-8 (pg/mL)	7.1	5.5–10.1	80.5	75.5–97.2	78.1	75.5–87.3	67.1	62.2–79.5	1.2	0.8–1.3
			p_1–2_ < 0.001		p_1–3_ < 0.001		p_1–4_ < 0.001		p_1–5_ = 0.01; p_2–5_ < 0.001; p_3–5_ < 0.001; p_4–5_ < 0.001	
IFNγ (U/mL)	6.6	4.2–7.4	2.9	2.2–4.0	3.2	2.3–4.8	4.4	3.3–6.9	12.4	10.5–14.9
			p_1–2_ < 0.001		p_1–3_ < 0.001		p_1–4_ < 0.001; p_2–4_ = 0.02		p_1–5_ < 0.001; p_2–5_ < 0.001; p_3–5_ < 0.001; p_4–5_ < 0.001	
TNF-β (pg/mL)	3.54	0.8–7.87	43.2	40.1–60.9	48.9	37.4–55.7	38.1	32.3–43.4	0.3	0.1–0.4
			p_1–2_ < 0.001		p_1–3_ < 0.001		p_1–4_ < 0.001		p_1–5_ < 0.001; p_2–5_ < 0.001; p_3–5_ < 0.001; p_4–5_ < 0.001	
IL-17A (pg/mL)	2.8	1.4–3.1	9.4	8.2–11.3	8.6	7.7–9.3	8.1	7.4–8.2	7.4	7.2–8.1
			p_1–2_ = 0.03		p_1–3_ = 0.03		p_1–4_ = 0.03		p_1–5_ = 0.03	
IL-6 (pg/mL)	3.2	1.3–4.8	60.5	59.1–67.8	68.4	66.3–72.1	75.1	72.4–81.1	88.1	79.4–97.1
			p_1–2_ < 0.001		p_1–3_ < 0.001		p_1–4_ < 0.001		p_1–5_ < 0.001	
IL-10 (pg/mL)	5.4	2.2–8.9	22.3	19.1–57.2	38.5	32.4–45.3	47.2	40.6–53.1	56.7	48.4–61.2
			p_1–2_ < 0.001		p_1–3_ < 0.001		p_1–4_ < 0.001		p_1–5_ < 0.001	
IL-4 (pg/mL)	4.1	3.6–5.5	86.8	76.8–103.5	91.4	73.2–112.3	93.3	68.6–122.1	2.2	1.1–3.4
			p_1–2_ < 0.001		p_1–3_ < 0.001		p_1–4_ < 0.001		p_1–5_ = 0.03; p_2–5_ < 0.001; p_3–5_ < 0.001; p_4–5_ < 0.001	

Note: p_1–2_—statistically significant differences between the indices of patients with gastric cancer (stage I) and the control group; p_1–3_—statistically significant differences between the indices of patients with gastric cancer (stage II) and the control group; p_1–4_—statistically significant differences between the indices of patients with gastric cancer (stage III) and the control group; p_1–5_—statistically significant differences between the indices of patients with gastric cancer (stage IV) and the control group; p_2–3_—statistically significant differences between the indices of patients with gastric cancer (stage II) and gastric cancer (stage I); p_2–4_—statistically significant differences between the indices of patients with gastric cancer (stage III) and gastric cancer (stage I); p_2–5_—statistically significant differences between the indices of patients with gastric cancer (stage IV) and gastric cancer (stage I); p_3–4_—statistically significant differences between the indices of patients with gastric cancer (stage III) and gastric cancer (stage II); p_3–5_—statistically significant differences between the indicators of patients with gastric cancer (stage IV) and gastric cancer (stage II); p_4–5_—statistically significant differences between the indicators of patients with gastric cancer (stage IV) and gastric cancer (stage III).

**Table 3 ijms-26-07609-t003:** Indicators of the prooxidant and antioxidant system in plasma in patients with gastric cancer at different stages of the disease. (Me, C_25_–C_75_, pm-u).

Indicators	Control Group N = 100 (1)		Stomach Cancer Stage I N = 75 (2)		Stomach Cancer Stage II N = 70 (3)		Stomach Cancer Stage III N = 69 (4)		Stomach Cancer Stage IV N = 67 (5)	
	Me	C_25_–C_75_	Me	C_25_–C_75_	Me	C_25_–C_75_	Me	C_25_–C_75_	Me	C_25_–C_75_
MDA, µmol/1 g protein	1.6	0.96–2.24	45.3	33.1–86.2	56.35	32.46–101.74	76.4	69.2–123.3	91.3	82.4–167.1
			p_1–2_ < 0.001		p_1–3_ < 0.001		p_1–4_ < 0.001		p_1–5_ < 0.001; p_2–5_ = 0.03	
SOD, units/min/1 g protein	204.41	151.05–250.32	231.2	176.1–256.9	235.2	133.7–462.27	174.1	169.9–200.1	157.2	148.1–189.3
			p_1–2_ < 0.001		p_1–3_ < 0.001		p_1–4_ = 0.03		p_1–5_ < 0.001	
CAT, µmol/s/1 g protein	0.27	0.2–0.39	0.16	0.15–0.28	0.14	0.13–0.18	0.15	0.13–0.19	0.1	0.08–0.17
			p_1–2_ = 0.04		p_1–3_ = 0.04		p_1–4_ = 0.04		p_1–5_ = 0.003	
GST, mmol/min/1 g protein	41.3	37.7–42.64	72.86	46.32–80.4	83.5	79.3–110.6	95.1	76.9–91.1	103.4	79.3–110.9
			p_1–2_ < 0.001		p_1–3_ < 0.001		p_1–4_ < 0.001		p_1–5_ < 0.001	
GPO µmol/1 g protein	105.9	81.19–162.38	148.33	120.4–201.5	168.6	158.7–211.5	210.39	183.4–221.6	223.2	191.2–254.3
			p_1–2_ < 0.001		p_1–3_ < 0.001		p_1–4_ < 0.001; p_2–4_ = 0.03; p_3–4_ = 0.03		p_1–5_ < 0.001; p_2–5_ = 0.03; p_3–5_ = 0.03	
CP mg/L	192.5	157.5–227.5	368.1	276.3–400.7	375.8	282.9–826.06	389.8	280.1–412.5	400.1	299.1–453.1
			p_1–2_ < 0.001		p_1–3_ < 0.001		p_1–4_ < 0.001		p_1–5_ < 0.001	

Note: p_1–2_—statistically significant differences between the indices of patients with gastric cancer (stage I) and the control group; p_1–3_—statistically significant differences between the indices of patients with gastric cancer (stage II) and the control group; p_1–4_—statistically significant differences between the indices of patients with gastric cancer (stage III) and the control group; p_1–5_—statistically significant differences between the indices of patients with gastric cancer (stage IV) and the control group; p_2–3_—statistically significant differences between the indices of patients with gastric cancer (stage II) and gastric cancer (stage I); p_2–4_—statistically significant differences between the indices of patients with gastric cancer (stage III) and gastric cancer (stage I); p_2–5_—statistically significant differences between the indices of patients with gastric cancer (stage IV) and gastric cancer (stage I); p_3–4_—statistically significant differences between the indices of patients with gastric cancer (stage III) and gastric cancer (stage II); p_3–5_—statistically significant differences between the indicators of patients with gastric cancer (stage IV) and gastric cancer (stage II); p_4–5_—statistically significant differences between the indicators of patients with gastric cancer (stage IV) and gastric cancer (stage III).

## Data Availability

Data are available on request to the corresponding author.

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
