# Peer review of "Cytokine Regulation and Oxidative Stress in Helicobacter Pylori-Associated Gastric Adenocarcinoma at Different Stages: Insights from a Cross-Sectional Study"

_ijms, 2025, doi:10.3390/ijms26157609_

Round 1

Reviewer 1 Report

Comments and Suggestions for Authors

In this cross-sectional study, the authors describe the interplay between cytokines and oxidative stress factors/markers in patients with Helicobacter pylori-associated gastric adenocarcinoma. The authors provide comprehensive study across plasma cytokines such as IL-6 and IL-10 and oxidative stress markers such as SOD. I went through the paper and have the following points to comment:

  1. Points of major concern:
  2. One of the points of weakness in this study is the lack of details about the strain of H. pylori. This information is required to provide longitudinal data which is of great importance in affecting the progression of the disease. Additionally, the grading of the adenocarcinoma should be correlated to the strain and/or oxidative stress.
  3. The authors need to provide the p-values and how they were calculated (for example in table 1) in this study. This information is required for biostatisticians aiming at developing data for therapeutic approaches.
  4. Does the oxidative stress and/or cytokines could be used as diagnostic tools for patients with this type of carcinoma? And what is the possibility of differential diagnosis from other carcinoma types in general? The authors need to include this in their study.
  5. How did the authors choose the sample size (the 281 patients)? The authors need to provide in percentages whether this size is comparable to the real clinical situation or not.

  1. Points of minor concern:
  2. Using only 18 references is very narrow and provide no clear clues about the real situation. In addition, the references included should be as new as possible and not out-dated.
  3. Table 2 looks overcrowded to the way making it difficult to clearly understand it. I would recommend dividing the table into further small tables to facilitate understanding it.
  4. In line 51, what do the authors mean with activated immunocompetent cells? How could be the cells activated and at the same time immunocompetent? The authors need to mention exactly which cells do they mean.
  5. The authors need to clarify in detail which methods were used to measure the cytokines and whether these methods were validated or not. I think this should be a nice refinement for the data mentioned in this study.
  6. Line 35, please correct the way of writing H2O2.
  7. The authors need urgently to revise the paper. There are some sections written using a different font size such as the author contribution.

Author Response

Comments1: One of the points of weakness in this study is the lack of details about the strain of H. pylori. This information is required to provide longitudinal data which is of great importance in affecting the progression of the disease. Additionally, the grading of the adenocarcinoma should be correlated to the strain and/or oxidative stress.

Respons1: The lack of detailed information on the H. Pylori strain is due to the fact that in our study we did not perform H. Pylori genotyping, nor did we culture the bacteria from gastric biopsy samples. In our study, we determined the presence or absence of H. Pylori infection. Uninfected patients were not included in the study. Our study included patients who were found to be infected with H. Pylori. In the future, we plan to conduct a study that will include patients with and without H. Pylori infection, which may allow us to talk about the correlation of the degree of malignancy of adenocarcinoma with the strain and/or oxidative stress.

Comments2: The authors need to provide the p-values and how they were calculated (for example in table 1) in this study. This information is required for biostatisticians aiming at developing data for therapeutic approaches.

Respons2: The calculation of p values is described in materials and methods: Statistical data processing was performed using the Statistica 7.0 software package (StatSoft, USA). The analysis of the conformity of the distribution type of the feature to the normal distribution law was performed using the Shapiro-Wilk criterion. When describing the sample, medians (Me) and the interquartile range of percentiles (C25-C75) were calculated. The reliability of differences between the indicators of independent samples was estimated using the Mann-Whitney criterion (p<0.05). Multivariate regression analysis was performed using two predictors. According to Green's rule , a minimum sample size of 66 subjects (N ≥ 50 + 8 × m, where m = number of predictors) is required to detect medium effect sizes when testing the general model. Given our sample size, the analysis meets this recommended criterion.

Comments3: Does the oxidative stress and/or cytokines could be used as diagnostic tools for patients with this type of carcinoma? And what is the possibility of differential diagnosis from other carcinoma types in general? The authors need to include this in their study.

Respons3: At this point, it is too early to talk about the possibility of using oxidative stress and/or cytokines as diagnostic tools for patients with this type of carcinoma. More research on this topic is needed, as well as a larger sample of patients. If we consider the possibilities of differential diagnosis with other types of carcinoma in general, it is necessary to study oxidative stress and cytokines in other types of cancer and analyze the results using advanced analytical and statistical programs.

Comments4: How did the authors choose the sample size (the 281 patients)? The authors need to provide in percentages whether this size is comparable to the real clinical situation or not.

Respons4: Initially, we recruited 350 patients with gastric carcinoma. Patients without H. Pylori infection were excluded from this group. Thus, our study included 281 patients, which is 80% of the initially recruited material, which corresponds to the real clinical situation.

Points of minor concern:

Comments1:Using only 18 references is very narrow and provide no clear clues about the real situation. In addition, the references included should be as new as possible and not out-dated.

Respons1:Thank you for your comment on this matter. We have taken this into account and supplemented the list of references with fresh sources.

Comments3: In line 51, what do the authors mean with activated immunocompetent cells? How could be the cells activated and at the same time immunocompetent? The authors need to mention exactly which cells do they mean.

Respons3: Immunocompetent cells are immune system cells that are capable of participating in the development of an immune response, i.e. interacting with antigens and triggering the body's defenses. These include, in particular, lymphocytes (T cells and B cells) and macrophages. Activated immunocompetent cells are immune system cells that have been activated in response to an antigen (a foreign substance that causes an immune response). These cells begin to function actively, performing various immune functions, such as destroying infected cells or producing antibodies. Immunocompetent cells, such as T lymphocytes, B lymphocytes, and macrophages, are normally in an inactive state. After contact with an antigen, they are activated and become active. T lymphocyte activation: Antigen recognition: T cells, which have specific receptors on their surface, recognize the antigen presented on the surface of antigen-presenting cells. Costimulation: For a T cell to be fully activated, in addition to recognizing the antigen, costimulation is required, which is the interaction between molecules on the surface of the T cell and the antigen-presenting cell. Proliferation and differentiation: Once activated, T cells begin to actively divide, forming a clone of cells capable of performing a specific function. Some of these cells become effector T cells that perform functions (e.g., producing cytokines or destroying infected cells), while others are transformed into memory T cells. The role of activated cells in the immune response: Effector T cells: Participate in the destruction of infected cells, regulate the immune response, and suppress excessive immune reactions. Antibodies: Neutralize antigens and promote their removal. Memory T cells: Provide a faster and more effective immune response upon repeated encounter with the same antigen. In general, activated immunocompetent cells play a key role in the development of the immune response, protecting the body from infections and other pathological processes. But it is not possible to insert this section into the article, as it is out of line with the overall concept of the article.

Comments4:The authors need to clarify in detail which methods were used to measure the cytokines and whether these methods were validated or not. I think this should be a nice refinement for the data mentioned in this study.

Respons4: The method for measuring cytokines is described in detail in the Materials and Methods section. Enzyme-linked immunosorbent assay (ELISA) is a laboratory immunological method for the qualitative or quantitative determination of various low-molecular compounds, macromolecules, viruses, etc., based on a specific antigen-antibody reaction.

Comments5:Line 35, please correct the way of writing H2O2.

Respons5:Thank you for this comment, we have made corrections.

Comments6: The authors need urgently to revise the paper. There are some sections written using a different font size such as the author contribution.

Respons6:Thank you for this comment, we have made corrections.

Reviewer 2 Report

Comments and Suggestions for Authors

Presentation of the results and discussion should be improved. Abbreviation used in this manuscript at times are confusing. It will be helpful to add a section on abbreviation used.

Discussion ends abruptly without any conclusions, may be it should be rewritten.

Some sentences are very long, loses its meaning.

Comments on the Quality of English Language

Presentation of the manuscript should be improved. 

Author Response

Comments: Presentation of the results and discussion should be improved. Abbreviation used in this manuscript at times are confusing. It will be helpful to add a section on abbreviation used. Discussion ends abruptly without any conclusions, may be it should be rewritten. Some sentences are very long, loses its meaning.

Respons: Thank you for your contribution to improving our article. We do not quite understand what you mean by improving the presentation of results and discussion. We do not think it is appropriate to make a whole section devoted to abbreviations. All abbreviations are written in the text as they are mentioned.